# Polymorphisms in *ARNTL/BMAL1* and *CLOCK* Are Not Associated with Multiple Sclerosis in Spanish Population

**DOI:** 10.3390/biology11101417

**Published:** 2022-09-28

**Authors:** Isabel de Rojas, César Martin-Montero, Maria Fedetz, Adela González-Jiménez, Fuencisla Matesanz, Elena Urcelay, Laura Espino-Paisán

**Affiliations:** 1Genetics and Molecular Bases of Complex Diseases Laboratory, Instituto de Investigación Sanitaria del Hospital Clínico San Carlos (IdISSC), 28040 Madrid, Spain; 2Department of Cell Biology and Immunology, Instituto de Parasitología y Biomedicina “López Neyra”, Consejo Superior de Investigaciones Científicas (IPBLN-CSIC), 18016 Granada, Spain

**Keywords:** genetic association, circadian rhythm, multiple sclerosis, autoimmune disease

## Abstract

**Simple Summary:**

Autoimmune diseases such as multiple sclerosis develop from an undefined mixture of environmental factors interacting with dozens to hundreds of genetic variations that confer risk. In the particular case of multiple sclerosis, shift work at a young age has been related to a higher risk of developing the disease. Shift work may alter the circadian rhythm, a series of complex molecular signals that inform our body of the day–night cycles. Circadian rhythm could be altered by external signals (longer days or nights, jet lag), but it is also genetically regulated through two master genes: *CLOCK*and *ARNTL/BMAL1*. A previous study in a population of Slavic origin found two genetic associations with multiple sclerosis risk in these genes. However, these genetic association studies require replication in independent populations to verify the true nature and magnitude of the effects on disease development. We replicated the study in an independent population and found no association of the two polymorphisms with multiple sclerosis.

**Abstract:**

Disrupted circadian cycle has been reported in multiple sclerosis (MS). Previous genome-wide association studies (GWAS) singled out over 230 variants associated with MS. A study performed in a Slavic population identified two new single nucleotide polymorphisms (SNPs), rs6811520 (*CLOCK*) and rs3789327 (*ARNTL/BMAL1*), associated with MS risk. However, these regions that codify the capital regulators of circadian rhythm had not been linked to the disease before, so replication in independent populations is warranted to ascertain possible geographical differences. Our aim was to replicate the associations reported in the *ARNTL/BMAL1* and *CLOCK* genes in a Spanish cohort with a maximum of 974 MS patients and 626 controls. In this study, 956 MS patients and 612 controls were successfully genotyped for rs6811520 and 943 MS patients and 598 controls for rs3789327.Clinical variables (age at disease onset, EDSS, or relapses) were collected in a maximum of 549 patients. No statistically significant differences were found between cases and controls for the analyzed SNPs, even after stratifications by sex, clinical form, or HLA-DRB1*15:01 status. No influence of the SNPs was found on age at disease onset, EDSS, or annual relapse rate at 5 years after onset. In conclusion, our study does not replicate the associations observed in the previously investigated Slavic population.

## 1. Introduction

Multiple sclerosis is a chronic, inflammatory, and demyelinating disease affecting the central nervous system. To date, it is one of the primary neurological causes of physical disability in young adults, with a growing prevalence in the past few decades [1]. The underlying cause is proposed to be an interaction between yet-to-be-defined genetic and environmental factors. Among the latter, a north–south gradient of MS prevalence has been consistently observed, with higher prevalence in northern countries and lower in those closer to the equator [2]. A higher incidence of MS has also been reported in shift workers, especially when shift work is performed at a young age [3].

Shift work, day–night duration, and low or uneven exposure to sunlight disrupt the circadian rhythm. This triggers a biochemical signaling cascade that informs the body of the day–night cycle and regulates several physiological processes, from hormone secretion to metabolism and sleep. Alterations in the circadian rhythm have been associated with autoimmune diseases [4]. Shift work has been reported as a susceptibility factor for the onset of MS [3], and links between the immune system and circadian rhythm regulation have been described in rheumatoid arthritis [5]. Therefore, the circadian rhythm constitutes an interesting objective to explore new genetic markers in MS and other autoimmune diseases. This cycle is mainly regulated by two genes: *ARNTL/BMAL1* and *CLOCK*. The products of these genes configure a heterodimer that regulates the genes *PER* and *CRY* in a feedback loop which indirectly influences several regulatory cascades. Thus, *ARNTL* and *CLOCK* genes are the master regulators in the human circadian cycle [4]. 

A recent case-control study performed in a Slavic population (Croatia, Serbia, and Slovenia) reported the association with MS risk of two single nucleotide polymorphisms (SNPs) [6]: rs3789327 in *ARNTL* (OR = 1.67 [1.35–2.07], *p* = 0.0001) and rs6811520 in *CLOCK* (OR = 1.40 [1.13–1.73], *p* = 0.002). The researchers performed tagging with four SNPs in each gene in a maximum of 900 patients and 1024 healthy controls. Interestingly, neither the reported SNPs nor others in the genetic regions surrounding *ARNTL* and *CLOCK* have been found associated with MS in previous genome-wide association studies (GWAS) [7]. This raises questions regarding insufficient coverage of the region in GWAS studies or specific characteristics of the patients recruited by Lavtar et al. that could be behind the differential association found. Therefore, in this study, we aimed to replicate the findings reported in the Slavic population in another well-characterized Caucasian population of a different origin.

## 2. Materials and Methods

### 2.1. Patients

A maximum of 974 MS patients (66.3% women) were recruited at Hospital Clínico San Carlos (Madrid, Spain). Patients were diagnosed according to the MacDonald criteria [8]. Concordantly, 626 healthy controls (60.2% women) were recruited among blood donors without personal or family history of autoimmune diseases. The following clinical data were collected for a maximum of 549 of the 974 patients (Table 1): form of MS, age at disease onset, and indicators of progression such as the Extended Disability Status Scale (EDSS) and annual relapse rate (ARR) at 5 years after onset. Additionally, the HLA-DRB1*15:01 status was determined for patients and controls (35% vs. 12.5% carriers of HLA-DRB1*15:01). Concerning the clinical form, 79.8% of the patients had relapsing remitting multiple sclerosis (RRMS), 15.5% had secondary progressive MS (SPMS), 3.3% had primary progressive MS (PPMS), and 1.6% had clinically isolated syndrome (CIS) or radiologically isolated syndrome (RIS) at the moment of collection. The clinical characteristics of these patients are summarized in Table 1.

Patients and controls were recruited after giving written informed consent, and the study was conducted according to the Declaration of Helsinki and approved by the Ethics Committee of Hospital Clínico San Carlos (protocol code 16/211-E, date of approval: 20 April 2016).

### 2.2. Genotyping

Patients and controls were genotyped with Taqman technology (Applied Biosystems, Foster City, CA, USA) for rs6811520 in *CLOCK* (catalogue number C__31137409_30), rs3789327 in *ARNTL* (catalogue number C___2160503_20), and rs3135388 (catalogue number C__27464665_30) for the HLA-DRB1*15:01 allele in a 7900HT Fast Real-Time PCR system. Briefly, 30 ng of genomic DNA mixed with 0.045 μL of the Taqman probe in study, 1.3 μL of Genotyping Master Mix (4371355, Applied Biosystems, Foster City, CA, USA), and 2.5 μL of distilled water to a total volume reaction of 5 μL were run in RT-PCR 7900HT equipment with the following protocol: 2 min at 50 °C for AmpErase activation, 10 min at 95 °C for DNA denaturation, and 40 cycles of denaturation and synthesis (15 s at 92 °C for denaturation of DNA and 1 min at 60 °C for hybridization and synthesis). Allelic discrimination was performed with SDS v2.4.1 software and genotype assignment was confirmed by reviewing the RT-PCR spectra of each sample. The genotyping call rate was above 95% in both groups. 

### 2.3. Statistical Analysis

Differences in allele, genotype, and carrier frequencies were calculated by contingency tables and chi-square tests. Statistical analyses stratified by sex, clinical form, and HLA-DRB1*15:01 status were performed. Associations were estimated using ORs with 95% confidence intervals in all cases. Normality was assessed in quantitative variables using the Kolmogorov–Smirnov test. Age at disease onset, EDSS, and ARR at five years after disease onset stratified by genotype in the studied polymorphisms were analyzed with Kruskal–Wallis (for genotype analyses) and U Mann–Whitney tests (for carrier analysis). Logistic regressions adjusting the analyses by sex, clinical form, and HLA were also performed for carriers of each SNP. In each carrier analysis, the association reported in the Lavtar et al. [6] study was selected as the risk genotype(s). Power calculation and statistical analyses were carried out using GRANMO v7.12 (Barcelona, Spain, https://www.imim.es/ofertadeserveis/software-public/granmo/, last accessed 17 July 2022), Epidat v3.1 (Santiago de Compostela, Spain), and SPSS v.15.0 software (Chicago, IL, USA). For comparison purposes, the data from the Lavtar et al. study [6] (total MS patients and controls, and genotype absolute frequencies in each SNP) were extracted from the information in the original paper, and all the statistics were recalculated.

## 3. Results

In the preliminary quality control analysis, both patients and controls followed the allele and genotype frequencies expected for the Hardy–Weinberg equilibrium.

No statistically significant differences were observed for rs6811520 (*CLOCK*) or rs3789327 (*ARNTL*) in the case-control analysis (Table 2) or in the analyses stratified by sex, clinical form, and presence of HLA-DRB1*15:01 (Appendix A). Logistic regression adjusted by these three variables (sex, clinical form, and HLA-DRB1*15:01) was also performed, and no association of the SNPs with risk of MS was found (Appendix A).

For both SNPs, the influence of the genetic content on age at disease onset and progression of disease (measured as EDSS and ARR at five years of disease onset) was also negative (Table 3).

## 4. Discussion

Lavtar et al. [6] reported the association of rs6811520 (*CLOCK*) and rs3789327 (*ARNTL/BMAL1*) with MS susceptibility in a Slavic population. These SNPs had not been previously associated with MS in GWAS or candidate gene studies, although they can be anticipated as interesting targets in MS pathogenesis. However, after careful examination of the original data presented in the article [6], we detected some technical problems concerning the rs6811520 (*CLOCK*) analysis that could explain the findings by Lavtar et al. in this region. Low success in the call rate was observed in the MS sample (79%). The control sample reached a higher call rate (91%), but was still below the quality standards required for this kind of study. The genotype and allele frequencies in control samples do not conform to those expected by Hardy–Weinberg equilibrium, and this should deter further analyses. The minor allele frequency (MAF) in the European population is 0.35 according to The 1000 Genomes project [9], similar to the 0.36 we found in our MS and control samples. However, the authors reported a minor allele frequency of 0.21 in controls. Population differences could be at play, but considering the low success in the call rate and the departure from Hardy–Weinberg equilibrium in controls, with an extreme difference in MAF between the authors’ results and those reported for the European population in public databases, it is arguable that these technical problems are behind the reported genetic association. In our replication study, both patient and control samples complied with the frequencies expected by Hardy–Weinberg equilibrium, our call rate is over 95% as perceptive, and the MAF in our control sample mirrors that in public databases. These quality control steps support the absence of technical errors in the design and execution of the study. Our sample size renders over 80% statistical power to detect the OR reported by the previous study, and nonetheless, we did not find statistically significant differences in our Spanish cohort.

A recent study in an Iranian population reported an association of rs6811520 (*CLOCK*) with MS [10]. However, the study apparently also suffers from technical errors and a low sample size. The authors studied 100 MS patients and 100 controls, but the genotypic frequencies of the SNP in controls did not adjust to Hardy–Weinberg equilibrium, with a marked lack of heterozygotes. In fact, the authors reported the association of the heterozygotes with an increased risk of MS, but did not discuss the biological significance of such an association that does not conform to additive or codominant models of inheritance.

Concerning the analysis of rs3789327 in the *ARNTL*gene, the call rate reported by Lavtar et al. is over 95%, the control population complies with Hardy–Weinberg expectations, and the MAF in controls is concordant with that reported in public databases for the European population. Our study reaches over 80% statistical power to detect the association reported in the previous study, but still, we do not find a significant association of this SNP. However, a milder effect could still exist. Although our study has enough statistical power to detect the effects described by Lavtar et al. in the original study, it is not powered to detect ORs of 1.10–1.15 that are usually found in MS genetic associations. 

The *ARNTL/BMAL1* region has shown a borderline association with depression in MS through a GWAS that studied the genetic factors of depression in an MS cohort [11]. The study reported the association of rs10832000 in 182 MS patients with depression and 1180 individuals with MS and no depression, but this polymorphism shows weak linkage disequilibrium (D’ = 0.048, R^2^ = 0.001) with rs3789327, the one studied in the Slavic population. Other genome-wide association studies performed in MS have not reported any association in the *ARNTL* gene [7], even though specific studies have focused on alterations in the circadian rhythm and susceptibility to neurological diseases [12]. The association reported by Lavtar et al. could reflect population differences or else a spurious association, and warrants replication in an independent cohort with the same ethnic origin to finally confirm or discard it.

In summary, the reported associations in *CLOCK* and *ARNTL*genes were not replicated in an independent Caucasian population of Spanish origin.

## Figures and Tables

**Table 1 biology-11-01417-t001:** Clinical characteristics of the recruited patients. For age at onset, Extended Disability Status Scale (EDSS), and annual relapse rate (ARR), the mean and standard deviation (in parenthesis) are presented. Only patients with available data in each category are included. Age at onset is measured in years. EDSS is a disability scale from 0 to 10, with 0 being no disability. ARR is the number of relapses corrected by a given time period (5 years after onset). The percentage of the different clinical forms is also presented in rows 4–7. Mean and standard deviation are presented for age at onset, EDSS, and ARR; percentage over the total population with confirmed clinical form is presented for CIS (clinically isolated syndrome), RIS (radiologically isolated syndrome), relapsing–remitting, secondary progressive, and primary progressive forms. All characteristics are calculated for the total sample, and for female and male patients.

	N	Overall Patients	N	Female	N	Male
Age at onset	506	29.29 (8.40)	351	29.22 (8.52)	155	29.46 (8.15)
EDSS at 5 year review	243	1.80 (1.52)	166	1.67 (1.47)	77	2.10 (1.61)
ARR during the first 5 years	309	0.89 (0.56)	210	0.87 (0.55)	99	0.92 (0.58)
CIS or RIS	9	1.6%	4	1.1%	5	2.7%
Relapsing remitting	437	79.8%	299	81.9%	138	75.0%
Secondary progressive	85	15.5%	50	13.7%	35	19.0%
Primary progressive	18	3.3%	12	3.3%	6	3.3%

**Table 2 biology-11-01417-t002:** Genotype distribution and case-control association study of *CLOCK* rs6811520 and *ARNTL* rs3789327. Odds ratio with 95% confidence intervals (OR 95% CI) are included. For the genotype study, *p* values correspond to the comparison of the three genotypes between cases and controls in 3 × 2 contingency tables, and ORs were calculated taking the TT genotype as a reference.

	MS		Controls		*p*	OR (95% CI)
	N	%	N	%		
*CLOCK* rs6811520	956		612			
TT	394	41.21	245	40.03	0.65	Reference
CT	433	45.29	282	46.08		0.95 (0.76–1.18)
CC	129	13.49	85	13.89		0.94 (0.68–1.29)
CC vs. Carrier of T	827	86.51	527	86.12	0.82	0.97 (0.72–1.30)
*ARNTL* rs3789327	943		598			
TT	213	22.59	151	25.25	0.17	Reference
TC	487	51.64	307	51.34		1.12 (0.87–1.44)
CC	243	25.77	140	23.41		1.23 (0.91–1.65)
CC vs. Carrier of T	700	74.23	458	76.59	0.30	1.13 (0.89–1.44)

**Table 3 biology-11-01417-t003:** Influence of CLOCK rs6811520 and ARNTL rs3789327 on age at onset and prognosis of MS. Age at onset is presented in years. Annual relapse rate (ARR) is calculated as the number of relapses in a given period (5 years after onset) divided by the time period. The Extended Disability Status Scale (EDSS) measures disability on a scale from 0 to 10, with 0 being no disability. Median and interquartile range (i.r.) are presented for all variables.

	Age at Onset			ARR 5 Years			EDSS 5 Years		
	N	Median (i.r.)	*p*	N	Median (i.r.)	*p*	N	Median (i.r.)	*p*
*CLOCK* rs6811520	493			301			236		
TT	210	28 (23–33)	0.21	128	0.8 (0.4–1.2)	0.88	103	1.5 (0.0–2.5)	0.49
TC	222	29 (23–36)		139	0.8 (0.6–1.2)		111	1.5 (1.0–3.0)	
CC	61	29 (24–35)		34	0.7 (0.6–1.2)		22	2 (1.5–3.5)	
CC vs. Carrier of T	432	28 (23–34)	0.50	267	0.8 (0.4–1.2)	0.79	214	1.5 (1–2.5)	0.41
*ARNTL* rs3789327	487			299			237		
TT	100	27 (23–32)	0.43	64	0.8 (0.6–1.2)	0.55	52	1.5 (1.0–3.0)	0.38
TC	259	28 (23–34)		155	0.8 (0.4–1.2)		122	1.5 (1.0–3.0)	
CC	128	29 (23–35)		80	0.6 (0.6–1.0)		63	1.5 (0–2.5)	
CC vs. Carrier of T	359	28 (23–34)	0.61	219	0.8 (0.5–1.2)	0.64	174	1.5 (1.0–3.0)	0.18

## Data Availability

Raw data can be accessed and downloaded via the “RepositorioInstitucional de la Consejería de Sanidad de la Comunidad de Madrid” (https://hdl.handle.net/20.500.12530/54675).

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
