# Peer review of "Polymorphisms in ARNTL/BMAL1 and CLOCK Are Not Associated with Multiple Sclerosis in Spanish Population"

_biology, 2022, doi:10.3390/biology11101417_

Round 1

Reviewer 1 Report

This study aimed at replicating in a Spanish cohort of 974 MS patients and 626 controls, the associations for rs6811520 (CLOCK) and rs3789327(ARNTL/BMAL1) polymorphisms with a higher risk of Multiple Sclerosis reported by Lavtar P. et al. Plos One 2018.

Some flaws have been noticed. Here below my comments:

The Authors declare that the analysis has been performed on 974 MS patients, but in Table 1 there are only 506 individuals. What about the other 468 MS patients? Why these patients are not mentioned? Paragraph 2.1 refers to percentages based on 506 individuals which is not correct.

Table 3 shows that genotyping for rs3789327was performed on 943 MS patients while rs6811520 was performed on 956 MS patients, instead of 974. Why Authors did not consider only MS patients with both genotypes?

Statistics: U Mann-Whitney test is a nonparametric test used on two independent groups therefore it cannot be used for the analysis of polymorphisms based on continuous or ordinal variables such as “age at disease onset, EDSS or Annual relapse rate at five years after disease onset”. Please use the proper statistical analysis.  

Author Response

This study aimed at replicating in a Spanish cohort of 974 MS patients and 626 controls, the associations for rs6811520 (CLOCK) and rs3789327(ARNTL/BMAL1) polymorphisms with a higher risk of Multiple Sclerosis reported by Lavtar P. et al. Plos One 2018.

Some flaws have been noticed. Here below my comments:

The Authors declare that the analysis has been performed on 974 MS patients, but in Table 1 there are only 506 individuals. What about the other 468 MS patients? Why these patients are not mentioned? Paragraph 2.1 refers to percentages based on 506 individuals which is not correct.

We have carefully rewritten the description of the studied population for the sake of clarity and we thank the reviewer for noticing it was confusing. The number of patients with confirmed MS diagnosis used for genotyping was 974, but we only had additional clinical data (age at onset, EDSS, ARR) of some of them. This cohort was recruited retrospectively from the Multiple Sclerosis Unit at Hospital Clínico San Carlos, which is a reference unit and receives patients from different parts of Spain and in different stages of the disease. Part of them had not been followed in the unit, therefore some data as EDSS or ARR at 5 years after onset were not available. That is the reason why the case-control study, which aims at replicating the findings by Lavtar et al, includes more patients than the associations between clinical characteristics and the polymorphisms studied.

Table 3 shows that genotyping for rs3789327 was performed on 943 MS patients while rs6811520 was performed on 956 MS patients, instead of 974. Why Authors did not consider only MS patients with both genotypes?

The SNPs studied are located in different genes and different chromosomes (ARNTL in chromosome 11, CLOCK in chromosome 4). No evidence of genetic interaction between them exists in the literature or in our own data. Therefore, we decided to analyze every SNP on its own with the total amount of patients genotyped for each case. Selecting only the patients that were successfully genotyped for both SNPs would lower the amount of data analyzed without adding any advantage.

Statistics: U Mann-Whitney test is a nonparametric test used on two independent groups therefore it cannot be used for the analysis of polymorphisms based on continuous or ordinal variables such as “age at disease onset, EDSS or Annual relapse rate at five years after disease onset”. Please use the proper statistical analysis.  

We have modified the description of these analyses in the Statistical analysis section (2.3), as the previous description might be confusing. To study the influence of the genetic component in clinical variables, we compared the data of each variable (age, EDSS, ARR) between the group of homozygotes for the risk allele and the carriers of the opposite allele. The variables age at disease onset, annual relapse rate and EDSS are quantitative. The U Mann-Whitney test is used to compare the data of a quantitative variable that does not follow the normal distribution in two independent groups. In our case, these groups would be defined by the genotype carried. The genotype groups are independent, the fact that a person has a certain genotype and a certain age at onset does not modify or influence the genotype or age at onset of other members of the group. In each case, we compare a quantitative variable that does not follow the normal distribution in two independent genotype groups. We have consulted the Statistics Unit of the hospital, and, in our experience and with their knowledge, the U Mann-Whitney would be the recommended test to compare these variables between the genotype groups. However, we are open to any suggestions and to perform other analyses.

Reviewer 2 Report

Review Polymorphisms in ARNTL/BMAL1….’ By Rojas-Pablo and coworkers 

This review report concerns a brief report by Rojas-Pablo on polymorphisms in master genes found in a previous GWAS analysis. The authors show that the polymorphisms are not associated with multiple sclerosis in a Spanish cohort in a scientifically sound manner. 

The paper is very well written, and the information presented is clear. 

Some minor issues that should be considered or added are mentioned below. 

Simple summary, line22, remove ‘of’ in ‘these genetic association studies require of replication…’. 

Genotyping, lines 102 and further, since the analysis is at the heart of this report, I would include some details on the cycling conditions and give the probes that were used. I assume that C__27464665_30 is a code for the probe used. Better give a reference on where to find the sequence or give the sequences (or the kit used) here. I realize that the values indicate the soundness of the assays performed, but some details in the methods should not be omitted. 

A question on the setup side of the study: Lavtar and coworkers present the age of the cohort (and s.d. values, table 1 in their paper). The authors were looking at patients ranging from approximately 30 to 50 years, if I am correct. Is this comparable to the cohort analysed in your studies? 

Lines 184, discussion, would the association really be a consequence of ‘population bottleneck’? The region is quite large, and the cohort of patients was established by collaborating genetic centers in the region (not a single center, so the population may not be that homogeneous). 

Recently, a paper came out that partially supported the findings of Lavtar in an Iranian cohort (although smaller in size). Details of the methods and description of the study look fine to me, and I would certainly include this paper in the discussion – it does add to the information presented in this study. See also Abbasi, Trends in Medical Sciences 2, July 2022.

Author Response

This review report concerns a brief report by Rojas-Pablo on polymorphisms in master genes found in a previous GWAS analysis. The authors show that the polymorphisms are not associated with multiple sclerosis in a Spanish cohort in a scientifically sound manner. 

 The paper is very well written, and the information presented is clear. 

 Some minor issues that should be considered or added are mentioned below. 

Simple summary, line22, remove ‘of’ in ‘these genetic association studies require of replication…’. 

We appreciate the kind comments of the reviewer and the suggestions provided which helped to improve the final version of our manuscript. The phrase has been modified.

Genotyping, lines 102 and further, since the analysis is at the heart of this report, I would include some details on the cycling conditions and give the probes that were used. I assume that C__27464665_30 is a code for the probe used. Better give a reference on where to find the sequence or give the sequences (or the kit used) here. I realize that the values indicate the soundness of the assays performed, but some details in the methods should not be omitted. 

We have included more details on the PCR conditions used. We used Taqman probes, from Applied Biosystems, to genotype both SNPs. Briefly, this technology consists of a preformed set of fluorescence-labeled probes, one for each allele of the SNP in study, and two primers to amplify the surrounding area. The probes are marked with fluorescent labels (one for each allele) and a quencher that hampers the emission of fluorescence. If the probe is compatible with the sequence in the subject of study, the fragment surrounding the SNP will be amplified and the probe degraded by the polymerase, releasing the fluorescent label to the PCR medium. Thus, by detecting the type of fluorescence that has been released, we can determine which allele or alleles were present in the subject of study. It is a very robust technique, quite reliable and easy to execute, that has been used in the last 15 years for SNP genotyping and gene expression. The sequences of the primers and probes are proprietary and we have not access to them or the capability to publish them, but the probes can be purchased at Thermo Life Sciences along with the Master Mix required (we have included the reference of this reagent) and the assays can be performed in an RT-PCR equipment.

A question on the setup side of the study: Lavtar and coworkers present the age of the cohort (and s.d. values, table 1 in their paper). The authors were looking at patients ranging from approximately 30 to 50 years, if I am correct. Is this comparable to the cohort analysed in your studies? 

Yes, it is. We do not have the age at sample extraction for all the subjects that participated in this study, since part of the DNA collection counts with limited clinical data besides a positive MS diagnosis, but the age at sample extraction in most of the population (including controls) ranged from 30 to 55 years. Moreover, considering that this is a genetic study and that Lavtar et al reported associations with risk to develop MS, the age of the cohort at the time of sampling would be irrelevant considering that the genetic sequence does not change with age. The only necessary criteria for inclusion in the study group is a positive MS diagnosis.

Lines 184, discussion, would the association really be a consequence of ‘population bottleneck’? The region is quite large, and the cohort of patients was established by collaborating genetic centers in the region (not a single center, so the population may not be that homogeneous). 

We have substituted the previous “could reflect a population bottleneck (Caucasians of Slavic origin from Serbia, Croatia and Slovenia)” for a more general “could reflect population differences”.

Recently, a paper came out that partially supported the findings of Lavtar in an Iranian cohort (although smaller in size). Details of the methods and description of the study look fine to me, and I would certainly include this paper in the discussion – it does add to the information presented in this study. See also Abbasi, Trends in Medical Sciences 2, July 2022.

We have read the study and we thank the reviewer for pointing it. We have included a brief discussion of the study in the Discussion section. However, we are afraid that it also apparently contains important errors that make the results unreliable. Letting apart the small size of the cohort, the biggest problem is that, according to the data presented in the analysis of rs6811520 in table 2, the genotypic frequencies in controls and MS patients are not in Hardy-Weinberg equilibrium. The frequencies reported by the authors in controls are: TT=44%, TC=17% and CC=39%, but according to Hardy-Weinberg they should be: TT=27.5%, TC=49.8% and CC=22.5%. In fact, the excessively low frequency of heterozygotes observed in controls (17%) should always be a matter of concern as, according to the Hardy-Weinberg equation, in almost any genetic distribution the heterozygotes are the most frequent or second most frequent category. Consulting the data of the 1000 Genomes Project, the minor allele for rs6811520 is allele T with a general MAF=0.25, but in their study, T is the major allele with a frequency of 0.54, and the authors do not explain or discuss this huge discrepancy.

The authors report the association of the heterozygote CT with higher risk of multiple sclerosis, but they do not explain the biological significance of this result,which does not fit with additive or codominant models. In our knowledge, there has not been any single description of such an association in genome association studies. 

Round 2

Reviewer 1 Report

Personally, when I read an article I expect to find what is reported in the abstract and I expect coherence along the paper. The current analysis has not been performed on 974 MS patients and 626 healthy controls (see table 2). It does not matter that the two SNPs are not related. Since there are difficulties in retrieving additional clinical data (which is an important flaw because it allows the following statistical analysis with almost half of the cohort), at least the Authors should try to be more coherent in presenting data.

Regarding the statistical analysis: which normality test has been performed on “age at disease onset, EDSS or Annual relapse rate at five years after disease onset”? If data are not normally distributed they should be presented as median and interquartile ranges instead of mean and standard deviation.

Since the genotypes are three (TT, TC, CC), I would perform the Kruskal-Wallis test. Please in the paragraph “Statistical analysis” of Methods, insert also the stratified analysis and the logistic regression. Please add these results (instead of “results not shown” lines 137 and 139) as supplementary tables.  

Author Response

We thank the reviewer for the suggestions that have allowed us to improve the final presentation of our manuscript.

Personally, when I read an article I expect to find what is reported in the abstract and I expect coherence along the paper. The current analysis has not been performed on 974 MS patients and 626 healthy controls (see table 2). It does not matter that the two SNPs are not related. Since there are difficulties in retrieving additional clinical data (which is an important flaw because it allows the following statistical analysis with almost half of the cohort), at least the Authors should try to be more coherent in presenting data.

It is standard practice in genetic association studies to report the total amount of subjects used for genotyping in the abstract, and the total amount that were successfully genotyped for each genetic variation along the manuscript. Both data allow the reviewers and readers to calculate the genotyping success rate (the percentage of the population that was successfully genotyped). This, and the compliance to Hardy-Weinberg equilibrium in the control population, are the first quality control checks that any genetic association study must pass to proceed with the statistical analysis. Therefore, we are following the standard practice in genetic association studies: we chose to report the number of subjects in the cohort used for genotyping in the abstract. As an example, genome wide studies that analyze thousands of independent SNPs also report the total size of the cohort, not the amount of patients that achieved a successful genotype in each SNP. Consider that a good quality study needs to reach at least a 95% genotype success call rate for each SNP. That means that the amount of patients successfully genotyped can only differ in a maximum of 5% from the total size of the cohort. A genotyping probe or a technique usually do not reach 100% call rate when using big samples (over 100 subjects). Even genome-wide association studies have to discard SNPs for not achieving a minimum of 95% genotyping call rate, and they have different numbers of subjects genotyped depending on the SNP studied. When performing a haplotype analysis, only those subjects with genotypes in all the SNPs are considered, as the reviewer suggested in the first revision. Since rs6811520 and rs3789327 are independent, to analyze each one individually with the maximum amount of genotypes obtained is appropriate. The opposite could lead to artificially lower genotyping call rates and bias in the posterior analysis.

The current study was performed in 974 patients and 626 controls, this is the maximum number of subjects that were used for genotyping. All of them were genotyped, and we obtained a successful genotyping in 956 patients and 612 controls in rs6811520 and 943 patients and 598 controls in rs3789327. Most of the subjects not successfully genotyped in one SNP were genotyped in the other.

We thank the reviewer’s comment that the different numbers in the samples used for clinical variables may be confusing and, following the suggestion, in an effort to be as clear as possible in the amount of patients used for each analysis in the manuscript, the numbers have now been included in the abstract, and they are detailed for each analysis in all the tables. Moreover, our data can be accessed without restrictions in a public repository for anyone to review or perform their own analyses (see “Data availability statement” in the manuscript and https://hdl.handle.net/20.500.12530/54675 for accessing the raw data).

 Regarding the statistical analysis: which normality test has been performed on “age at disease onset, EDSS or Annual relapse rate at five years after disease onset”? If data are not normally distributed they should be presented as median and interquartile ranges instead of mean and standard deviation.

We used the Kolmogorov-Smirnov test to check for normality in SPSS. We have also substituted the mean and standard deviation for median and interquartile range in table 3, as suggested by the reviewer.

Since the genotypes are three (TT, TC, CC), I would perform the Kruskal-Wallis test. Please in the paragraph “Statistical analysis” of Methods, insert also the stratified analysis and the logistic regression. Please add these results (instead of “results not shown” lines 137 and 139) as supplementary tables.  

As suggested, we have added the genotype comparison with Kruskall-Wallis test in both SNPs to table 3, along with the “CC vs. Carrier of T” comparison. We have also included the description of the logistic regression analysis in Methods. The wording of the already included stratified analysis was modified, for the sake of clarity.

As indicated, we have now included tables with all the results previously not shown in Supplementary file 1, which adds the logistic regression analysis adjusted by sex, clinical form of MS, and HLA status (performed with SPSS) and also the stratified analysis by sex, clinical form, and HLA status. For the latter analyses, we have included the genotype comparisons in 3x2 contingency tables, and the p-value of each comparison. This format seems the most informative to us, since any reader could easily take the data and perform carrier or allele analyses.

Regarding a p<0.05 in the sex stratification analysis in ARNTL rs3789327 (the only comparison where we included a carrier analysis),we do not mention it in the manuscript since this is not a solid result, as:

- A solid sex difference would require that the distribution of rs3789327 in the MS female group would be different from the Control female, but also from the MS male. In our case, the MS female and male groups are not statistically different, neither the female and male controls.

-The difference in our sex-stratified analysis would not even be a replication of the original study. They reported an increase in CC frequency in MS patients compared to controls and they did not detect anything in the analysis stratified by sex. The OR detected in our analysis in “CC vs Carrier of T” between MS female and Control female reflects a lower frequency of CC in the MS group compared to female controls and, therefore, a “protective effect” of the CC genotype (OR=0.72 (0.53-0.97)). Nonetheless, the effect described by Lavtar et al. in their population was the opposite.

-The application of Bonferroni correction would require that any statistically significant p could be multiplied by 2 (the number of SNPs analyzed), and remained significant. The comparison is borderline significant (p=0.04) and does not stand Bonferroni correction.

-Finally, the logistic regression analysis adjusted by sex, clinical form, and HLA status does not detect any association of the SNP with MS risk. If there were a solid sex effect, we should see it also in this analysis.

When performing stratified analyses, one must be very careful interpreting the data and consider only solid associations. The study groups are subgroups of the original sample and have lower statistical power to detect associations. Also, the number of statistical comparisons performed when stratifying by several variables increases the likelihood of obtaining a false positive result, the reason why it is important that the detected associations stand a statistical correction.